# COVID-19 vaccination and major cardiovascular and haematological adverse events in Abu Dhabi: retrospective cohort study

Marco A. F. Pimentel[1], Maaz Shaikh [1] ✉, Muna Al Safi[2], Yousuf Naqvi[2] & Shadab Khan[1] ✉

The widespread administration of COVID-19 vaccines has prompted a need to understand their safety profile. This investigation focuses on the safety of inactivated and mRNA-based COVID-19 vaccines, particularly concerning potential cardiovascular and haematological adverse events. A retrospective cohort study was conducted for 1.3 million individuals residing in Abu Dhabi, United Arab Emirates, who received 1.8 million doses of the inactivated BBIBP CorV (by SinoPharm) and mRNA-based BNT162b2 (Pfizer-BioNTech) vaccines between June 1, 2021, and June 30, 2022. The study's primary outcome was to assess the occurrence of selected cardiovascular and haematological events leading to hospitalization or emergency room visits within 21 days post-vaccination. Results showed no significant increase in the incidence rates of these events compared to the subsequent 22 to 42 days following vaccination. Analysis revealed no elevated risk for adverse outcomes following first (IRR 1·03; 95% CI 0·82-1·31), second (IRR 0·92; 95% CI 0·72-1·16) and third (IRR 0·82; 95% CI 0·66-1·00) doses of either vaccine. This study found no substantial link between receiving either mRNA and inactivated COVID-19 vaccines and a higher likelihood of cardiovascular or haematological events within 21 days after vaccination.

The first global rollout of coronavirus disease 2019 (COVID-19) vaccines began in December 2020. Multiple COVID-19 vaccines were authorized for use in Abu Dhabi (UAE) and made freely available to the population as early as September 2020. As a result, the United Arab Emirates achieved one of the highest vaccination rates globally with a coverage of 89.1% of the population by November 2021[1]. During this period of vaccination, the most widely administered COVID-19 vaccines, to date, include mRNA-based vaccines, that is, BNT162b2 (by Pfizer-BioNTech) and mRNA-1273 (Moderna) vaccines, the inactivated BBIBP-CorV (SinoPharm), ChAdOx1 (Oxford-AstraZeneca), and Sputnik V (or Gam-COVID-Vac). BNT162b2 and BBIBP-CorV accounted for the largest proportion of vaccines administered (>90%).

Notably, a variety of vaccine platforms have been developed and deployed, each with unique mechanisms of action and safety profiles[2,3]. While mRNA-based vaccines, such as those developed by Pfizer-BioNTech and Moderna, have been widely discussed and are recognized for their high efficacy, especially against the omicron variants[4], it is crucial to consider the safety and efficacy of other vaccine types[5]. In particular, inactivated vaccines, such as those predominantly used in the MENA region and China[6,7], represent a significant portion of the global vaccination effort. These vaccines, which use a traditional approach of employing killed viral particles to stimulate an immune response, have shown differing safety profiles compared to mRNA vaccines[8,9]. A recently-published systematic

[1]M42, Abu Dhabi, UAE. [2]Department of Health (DOH), Abu Dhabi, UAE. ✉e-mail: mshaikh@m42.ae; skhan@m42.ae

review and meta-analysis concluded that inactivated vaccines, such as BBIBP-CorV show high effectiveness against severe COVID-19 outcomes like admission to the Intensive Care Unit and death and suggests that further evidence on their safety is needed[10]. The differences in adverse event profiles and long-term safety data of these vaccines are an important aspect of the broader discussion on COVID-19 vaccine safety.

While these vaccines have proven their efficacy in clinical trials, many case reports and studies have reported increased risk of different cardiovascular and haematological (CVDH) adverse events such as thrombocytopenia, venous thromboembolism, myocardial infarction and stroke for multiple types of vaccines used globally[11–13]. There is limited evidence on the safety profile of these vaccines in real-world setting at a regional population level for risk of major CVDH events[14]. In addition, the ethnic diversity in the resident population and high vaccination rates in Abu Dhabi provide a unique opportunity to evaluate the safety profile of these vaccines in a diverse population in addition to providing important regional data[15].

In this work, we used a large database of electronic health records to investigate the associations between the first, second, and third doses of the commonly administered COVID-19 vaccines in the region and selected CVDH events resulting in hospitalization and emergency room (ER) admissions. We also assessed risks for the same outcomes following vaccination in younger persons (<40 years old), persons at higher risk of the outcomes of interest, and other stratification groups. Incidence rate ratios, the rate of hospital admission or ER visit in risk periods after vaccination relative to concurrent comparison periods, were estimated using a modified rapid cycle analysis methodology[16–19].

## Results

A total of 1,793,162 BBIBP-CorV or BNT162b2 vaccine doses were administered to 1,312,505 people 12 years or older between 1st May 2021 and June 2022 in Abu Dhabi as captured in the Malaffi database (Table 1). Individuals receiving either vaccine were, on average, of similar age, with less than one-third of the overall population ($n = 429488$, 32·7%) being 40 years or older. We note that individuals who were administered the third dose (of either vaccine) are older than those who received doses 1 and 2 (within our study period), as the third dose was administered primarily to higher-priority risk groups (including older people). The proportion of males among individuals who received the BBIBP-CorV vaccine is higher, and more UAE Nationals (in relative terms) received the BNT162b2 vaccine. Of the total number of individuals, 7766 (less than 1%) had at least one related health outcome of interest recorded in the one year prior the vaccine administration. In addition, we note that our study period began at time when the vaccine rollout was already in progress, which explains the high proportion of individuals that received the third dose having one dose only, in comparison to that of individuals who received the first two doses (Table 1).

During the study period there were 474 inpatient admissions or ER visits related to at least one of the outcomes of interest that occurred within the risk interval of 1 to 21 days following vaccination. Note that one individual may have experienced more than one event, after the various dose administrations. In our post-hoc power analysis, we assessed the statistical power of our study in detecting meaningful effects based on the observed incidence rate of the primary outcome and found that to detect a 20% increase in risk with 80% power at a significance level of 0.05, a sample size of approximately 2.9 million participants would be required. Sample sizes of approximately 1.3 million and 0.5 million participants would be necessary to detect a 30% and 50% increase in risk, respectively. With our current sample size, our study demonstrates the ability to detect a risk increase of ~30%.

Admissions or ER visits related to acute myocardial infarction ($n = 212$) and non-haemorrhagic stroke ($n = 117$) and venous thromboembolism ($n = 114$) were the most prevalent outcomes of interest among the study population. Table 2 shows the number of individuals with the outcomes of interest.

### Primary outcome

Of the 1,312,505 vaccinated individuals included in our study, 4994 (0·38%) visited the ER or were admitted to hospital with at least one of the outcomes of interest (primary outcome) at any time in the study period; for 399 (0·03%) of these individuals, these events occurred in the 1–21 days post any dose of BBIBP-CorV or BNT162b2 vaccines (Table 2).

Tables 3 and 4 show the number of patients with primary outcome events (any of the outcomes of interest) in each exposure risk interval and the incidence rate ratios (IRRs) and corresponding 95% confidence intervals (CIs) for the primary outcome in the exposure risk intervals compared to the concurrent comparator period. The number of outcome events during the 21-day risk interval post either of the vaccines ranged from 131 (11·15 per 1,000,000 person-days) after dose 1, 122 (9·86 per 1,000,000 person-days) after dose 2 and 163 (15·10 per 1,000,000 person-days) after dose 3 (Table 3). The number of outcome events following each of the vaccines ranged from 9·38 per million person-days and 14·30 per million person-days for the BBIBP-CorV vaccine and from 9·29 per million person-days and 17·07 per million person-days for the BNT. In general, the (unadjusted) number of events (per million person-days, i.e., event rate) were higher in the risk interval when compared to that in the concurrent comparison interval, particularly following dose 3 of either vaccine.

Over the 1–21 days post-vaccination risk interval, we observed no association of the occurrence of the primary outcome with any of the doses of both vaccines (Table 3). There was no increased risk of the primary outcome during 1–21 days following the first (IRR 1·03; 95% CI 0·82, 1·31), second (IRR 0·92; 95% CI 0·72, 1·16) and third (IRR 0·82; 95% CI 0·66, 1·00) doses of either vaccine.

There was an increased risk of the primary outcome at 1–7 days following the second dose of BNT162b2 (IRR 1·47; 95% CI 1·02, 2·44). No increased risk of the primary outcome at 8–14 and 15–21 days following any of the exposures was observed.

### Subgroup analyses by outcome of interest

Of the 1,312,505 vaccinated individuals included in our study, 1486 (0·113%) were admitted to the hospital or visited the ER with a diagnosis of non-haemorrhagic stroke at any time in the study period (either before or after vaccination); for 103 (0·008%) of these individuals, the events occurred in the 1–21 days post any dose (i.e., risk interval) of either vaccine (Table 2). 462 (0·04%) individuals had a haemorrhagic stroke-related event at any time in the study period, and 35 (0·003%) of these had it in the risk interval. 1918 (0·146%) individuals had an event related to acute myocardial infarction at any time; 165 (0·013%) of these had the event in the risk interval. The number of individuals who had events related to myocarditis/pericarditis, pulmonary embolism, venous thromboembolism, and disseminated intravascular coagulation at any time in the study period were 171 (0·013%), 422 (0·032%), 1278 (0·097%) and 37 (0·003%), respectively; 15 (0·001%), 42 (0·003%), 103 (0·008%) and 2 (<0·001%), respectively, had the events within the 1–21 days post-vaccination risk interval.

Table 5 shows the IRRs for each individual outcome event of interest in the overall 1–21-day risk interval after each exposure of either BBIBP-CorV or BNT162b2 vaccines. No increased risk of each individual outcome-related events in the 1–21 days following dose 1, dose 2, and dose 3 of either vaccine was found.

### Subgroup analyses by age group and gender

Table S2 shows the IRRs for primary outcome events in the overall 1–21-day risk interval after each exposure by gender and in those aged under 40 years or 40 years and older.

**Table 1 | Baseline demographic characteristics of people receiving either BBIBP-CorV or BNT162b2 vaccines, in Abu Dhabi during the study period. Data are presented as column % (counts)**

| | BBIBP-CorV | | | BNT162b2 | | | Either vaccine |
|---|---|---|---|---|---|---|---|
| | Dose 1 | Dose 2 | Dose 3 | Dose 1 | Dose 2 | Dose 3 | Either dose |
| Total number of people | 317,962 | 383,060 | 369,841 | 298,322 | 275,311 | 148,666 | 1,312,505 |
| **Gender** | | | | | | | |
| Female | 27·3% (86,688) | 23·1% (88,296) | 14·8% (54,691) | 38·2 %(114,057) | 35·6 (97,984) | 34·0 (50,618) | 26·3 (345,722) |
| Male | 72·7 (231,175) | 76·9 (294,652) | 85·2 (315,037) | 61·7 (184,181) | 64·4 (177,254) | 65·9 (97,968) | 73·6 (966,359) |
| Unknown | <0·1 (99) | <0·1 (112) | <0·1 (113) | <0·1 (84) | <0·1 (73) | <0·1 (80) | <0·1 (424) |
| **Age** | | | | | | | |
| Median age (IQR) | 32 (25–40) | 33 (27–40) | 37 (31–45) | 32 (23–39) | 32 (24–40) | 39 (32–47) | 35 (27–42) |
| 12–17 years | 6·9 (21,784) | 2·6 (9850) | 0·3 (1024) | 17·1 (50,931) | 15·1 (41,535) | 0·6 (868) | 5·9 (77,876) |
| 18–39 years | 67·5 (214,571) | 70·4 (269,746) | 58·8 (217,415) | 58·0 (172,950) | 59·4 (163,507) | 52·0 (77,304) | 61·3 (805,141) |
| 40–64 years | 24·4 (77,716) | 26·0 (99,435) | 39·2 (144,989) | 23·7 (70,583) | 24·3 (66,822) | 44·5 (66,092) | 31·2 (409,235) |
| 65+ years | 1·2 (3891) | 1·1 (4029) | 1·7 (6413) | 1·3 (3858) | 1·3 (3447) | 3·0 (4402) | 1·5 (20,253) |
| **Country of origin (WHO region)** | | | | | | | |
| National | 11·7 (37,088) | 7·6 (29,281) | 10·5 (38,913) | 18·3 (54,670) | 15·7 (43,225) | 19·1 (28,359) | 13·0 (170,915) |
| GCC | 0·5 (1614) | 0·4 (1681) | 0·4 (1560) | 1·5 (4504) | 1·3 (3551) | 1·3 (1885) | 0·8 (10,617) |
| Africa | 5·2 (16,498) | 4·5 (17,176) | 2·0 (7351) | 3·9 (11,520) | 3·5 (9552) | 2·1 (3126) | 3·4 (44,730) |
| Americas | 0·6 (1864) | 0·4 (1408) | 0·4 (1317) | 1·3 (3875) | 1·5 (4150) | 2·2 (3302) | 0·9 (12,043) |
| Eastern Mediterranean | 36·6 (116,530) | 33·9 (129,758) | 23·9 (88,413) | 24·2 (72,084) | 23·7 (65,306) | 20·4 (30,381) | 27·4 (360,256) |
| Europe | 0·6 (2018) | 0·7 (2774) | 0·5 (1724) | 2·5 (7568) | 3·0 (8374) | 3·8 (5591) | 1·5 (20,336) |
| South-East Asia | 38·8 (123,220) | 46·2 (176,824) | 54·0 (199,837) | 40·8 (121,667) | 44·1 (121,303) | 34·8 (51,713) | 44·6 (585,413) |
| Western Pacific | 5·0 (15,965) | 5·4 (20,637) | 7·7 (28,334) | 6·4 (19,166) | 6·2 (16,974) | 15·1 (22,441) | 7·3 (96,030) |
| Unknown | 1·0 (3165) | 0·9 (3521) | 0·6 (2392) | 1·1 (3268) | 1·0 (2876) | 1·3 (1868) | 0·9 (12,165) |
| **Number of doses** | | | | | | | |
| One dose only | 28·5 (90,588) | 38·2 (146,183) | 95·4 (352,686) | 22·8 (67,959) | 15·4 (42,420) | 96·1 (142,899) | 64·2 (842,735) |
| **Health Outcomes of Interest in the 1 year preceding first vaccination** | | | | | | | |
| Previous outcomes of interest | 0·5 (1550) | 0·5 (1760) | 0·6 (2393) | 0·5 (1607) | 0·5 (1353) | 0·9 (1408) | 0·6 (7766) |
| No previous outcomes of interest | 76·0 (241,613) | 78·6 (300,895) | 86·1 (318,505) | 97·1 (289,639) | 97·3 (267,847) | 98·3 (146,168) | 86·7 (1,138,406) |
| No previous records | 23·5 (74,799) | 21·0 (80,405) | 13·2 (48,943) | 2·4 (7076) | 2·2 (6111) | 0·7 (1090) | 12·7 (166,333) |

**Table 2 | Number of outcomes and patients who experienced the individual outcomes in the 1–21 days following a first, second, or third doses of either COVID-19 vaccine**

| | Risk interval | | | Comparison interval | | |
|---|---|---|---|---|---|---|
| | Total number of events | Emergency events, *n* (% of events) | Number of patients, *n* (% of total count) | Total number of events | Emergency events, *n* (% of events) | Number of patients, *n* (% of total count) |
| Primary outcome | 474 | 179 (37·8%) | 399 (0·030%) | 532 | 222 (41·7%) | 446 (0·034%) |
| **Outcomes of interest** | | | | | | |
| Non-haemorrhagic stroke, NHS | 117 | 54 (46·2%) | 103 (0·008%) | 149 | 67 (45·0%) | 129 (0·010%) |
| Haemorrhagic stroke, HS | 38 | 25 (65·8%) | 35 (0·003%) | 35 | 24 (68·6%) | 32 (0·002%) |
| Acute myocardial infarction, AMI | 212 | 49 (23·1%) | 165 (0·013%) | 202 | 58 (27·9%) | 165 (0·013%) |
| Myocarditis/Pericarditis, M/P | 17 | 6 (35·3%) | 15 (0·001%) | 22 | 14 (63·6%) | 20 (0·002%) |
| Pulmonary embolism, PE | 49 | 24 (49·0%) | 42 (0·003%) | 50 | 24 (48·0%) | 45 (0·003%) |
| Disseminated intravascular coagulation DIC | 2 | 0 (0·0%) | 2 (<0·001%) | 2 | 0 (0·0%) | 2 (<0·001%) |
| Venous thromboembolism, VTE | 114 | 54 (47·4%) | 103 (0·008%) | 128 | 61 (47·7%) | 111 (0·008%) |

The number of outcome events (in person-days) was higher in individuals aged 40 years or older than that in those individuals aged under 40 years. This is expected in that the incidence of CVDH conditions is found to be higher in older individuals. In those aged under 40 years, the number of outcome events following either vaccine ranged from 4·37 per million person-days to 5·09 per million person-days, whereas in those aged 40 years or older the number ranged from 24·18 per million person-days to 31·34 per million person-days. We observed no increased risk of primary outcome events in the 1–21 days following dose 1, dose 2, and dose 3 of the BBIBP-CorV and BNT162b2 vaccines among those aged under 40 years, and among those aged 40 years or older.

**Table 3 | IRRs (incidence rate ratios) and 95% CI for the primary outcome in the predefined risk interval of 1–21 days after exposure to vaccination**

|  | Events in risk interval per 1 M pdays (n) | Events in comparison interval per 1 M pdays (n) | IRR (95% CI) | p-value |
|---|---|---|---|---|
| Either vaccine |  |  |  |  |
| Dose 1 | 11·15 (131) | 10·86 (213) | 1·03 (0·82–1·31) | 0·778 |
| Dose 2 | 9·86 (122) | 10·86 (213) | 0·92 (0·72–1·16) | 0·472 |
| Dose 3 | 15·10 (163) | 12·69 (319) | 0·82 (0·66–1·00) | 0·051 |
| BBIBP-CorV |  |  |  |  |
| Dose 1 | 12·90 (78) | 11·83 (139) | 1·20 (0·90–1·61) | 0·215 |
| Dose 2 | 9·38 (65) | 11·83 (139) | 0·81 (0·60–1·10) | 0·173 |
| Dose 3 | 14·30 (110) | 13·22 (224) | 0·83 (0·65–1·06) | 0·140 |
| BNT162b2 |  |  |  |  |
| Dose 1 | 9·29 (53) | 9·43 (74) | 0·76 (0·51–1·50) | 0·193 |
| Dose 2 | 10·13 (57) | 9·43 (74) | 1·22 (0·89–1·74) | 0·121 |
| Dose 3 | 17·07 (53) | 11·58 (95) | 0·70 (−0·73–1·01) | 0·052 |

Number of events in both risk and comparison intervals per million person-days (i.e., incidence rate, IR) is also shown as IR (n = number of events).
CI confidence interval, pdays person-days.

**Table 4 | IRRs (incidence rate ratios) and 95% CI for the primary outcome in the predefined risk intervals of 1–7, 8–14, and 15–21 days after exposure to vaccination**

|  | 1–7 days | | | 8–14 days | | | 15–21 days | | |
|---|---|---|---|---|---|---|---|---|---|
|  | Events in risk interval per 1 M pdays (n) | IRR (95% CI) | p-val | Events in risk interval per 1 M pdays (n) | IRR (95% CI) | p-val | Events in risk interval per 1 M pdays (n) | IRR (95% CI) | p-val |
| Either vaccine |  |  |  |  |  |  |  |  |  |
| Dose 1 | 10·73 (38) | 0·97 (0·67–1·41) | 0·876 | 11·31 (45) | 1·06 (0·75–1·48) | 0·749 | 11·34 (48) | 1·10 (0·80–1·52) | 0·552 |
| Dose 2 | 9·90 (43) | 1·12 (0·72–1·46) | 0·804 | 10·78 (45) | 1·07 (0·77–1·48) | 0·702 | 7·53 (34) | 0·76 (0·52–1·09) | 0·139 |
| Dose 3 | 13·11 (47) | 0·77 (0·56–1·07) | 0·117 | 15·82 (57) | 0·85 (0·63–1·14) | 0·276 | 16·35 (59) | 0·83 (0·62–1·12) | 0·223 |
| BBIBP-CorV |  |  |  |  |  |  |  |  |  |
| Dose 1 | 11·49 (21) | 1·06 (0·65–1·71) | 0·826 | 12·25 (25) | 1·10 (0·71–1·73) | 0·662 | 14·70 (32) | 1·28 (0·86–1·90) | 0·224 |
| Dose 2 | 8·30 (17) | 0·75 (0·45–1·25) | 0·269 | 13·12 (30) | 1·16 (0·77–1·73) | 0·483 | 6·93 (18) | 0·58 (0·35–0·96) | 0·033 |
| Dose 3 | 11·76 (30) | 0·75 (0·50–1·13) | 0·170 | 17·53 (45) | 1·03 (0·73–1·45) | 0·870 | 13·60 (35) | 0·78 (0·53–1·13) | 0·189 |
| BNT162b2 |  |  |  |  |  |  |  |  |  |
| Dose 1 | 9·92 (17) | 0·83 (0·45–1·54) | 0·559 | 10·33 (20) | 0·73 (0·41–1·29) | 0·274 | 7·78 (16) | 0·69 (0·38–1·25) | 0·221 |
| Dose 2 | 11·88 (26) | 1·47 (1·02–2·44) | 0·041 | 7·95 (15) | 0·63 (0·34–1·19) | 0·155 | 8·34 (16) | 1·00 (0·57–1·73) | 0·987 |
| Dose 3 | 16·46 (17) | 0·63 (0·36–1·12) | 0·115 | 11·59 (12) | 0·49 (0·26–0·91) | 0·023 | 23·18 (24) | 0·91 (0·57–1·45) | 0·686 |

CI confidence interval, pdays person-days, p-val p-value.

No association of primary outcome events was found in either men or women in the 1–21-day risk interval following either exposure (see Table S2 in supplementary materials).

**Subgroup analyses by occurrence of previous events**
We categorized individuals into high and low-risk groups, based on prior history of any of the outcomes of interest in the 12-month period preceding the exposure. 7766 (0·6% of all individuals) individuals had at least one event of any of the outcomes of interest recorded in the one year prior the vaccine administration, hence were considered as part of the high-risk group; whereas 1,138,406 (86·7%) did not have any occurrence of the selected outcome events in the year before vaccination and were assigned to the low-risk group. As expected, we observed that the rate of (primary) outcome events (per million person-days) was higher in the high-risk group, ranging from 433·33 to 735·96 post-exposure to either vaccine, compared to the low-risk group, which ranged from 7·54 to 13·03 (see Table S3 in supplementary materials). No increased risk of the primary outcome events in the 1–21 days following dose 1, dose 2, and dose 3 of either vaccine was found with-in each group.

## Discussion
This is one of the largest studies to date of CVDH outcomes after COVID-19 vaccination to be conducted in the Gulf and MENA region, one of the first studies to compare the risk of cardiac and haematological adverse events between different vaccine products, and the first to investigate the association between these outcome events and the BBIBP-CorV vaccine.

The findings of this study are relevant to clinicians and policymakers and may complement current surveillance and vaccine safety monitoring systems. There was an increase in the risk of CVDH adverse events with the first week of receiving the second dose of the BNT162b2 vaccine. In contrast, we found no evidence of an increase in the risk of any individual outcome following vaccination. Also, we found no association of adverse events in any of the subgroups considered in this study following exposures to either vaccine.

The association between COVID-19 vaccination and cardiac events using confirmed cases and hospital admissions with diagnoses of myocarditis or pericarditis (in particular) has been assessed in previous studies[20–22]. In addition, in response to concerns regarding associations between thromboembolic outcomes and COVID-19

**Table 5 | IRRs (incidence rate ratios) and 95% CI for each secondary outcome (or event) of interest in the predefined risk intervals of 1–21 days after exposure to either BBIBP-CorV or BNT162b2 vaccines**

| | Events in risk interval per 1 M pdays (n) | Events in comparison interval per 1 M pdays (n) | IRR (95% CI) |
|---|---|---|---|
| Non-haemorrhagic stroke | | | |
| Dose 1 | 3·15 (37) | 2·91 (57) | 0·98 (0·63–1·54) |
| Dose 2 | 3·08 (38) | 2·91 (57) | 1·14 (0·72–1·80) |
| Dose 3 | 3·15 (34) | 3·58 (90) | 0·67 (0·44–1·02) |
| Haemorrhagic stroke | | | |
| Dose 1 | 0·94 (11) | 0·77 (15) | 2·03 (0·84–4·92) |
| Dose 2 | 1·03 (13) | 0·77 (15) | 1·73 (0·82–3·65) |
| Dose 3 | 1·02 (11) | 0·76 (19) | 0·73 (0·33–1·60) |
| Acute myocardial infarction | | | |
| Dose 1 | 3·57 (42) | 3·62 (71) | 1·12 (0·74–1·67) |
| Dose 2 | 3·58 (45) | 3·62 (71) | 1·01 (0·69–1·48) |
| Dose 3 | 7·78 (84) | 4·77 (120) | 0·99 (0·73–1·34) |
| Myocarditis/Pericarditis | | | |
| Dose 1 | 0·68 (8) | 0·61 (12) | 0·48 (0·18–1·32) |
| Dose 2 | 0·40 (5) | 0·61 (12) | 0·35 (0·11–1·11) |
| Dose 3 | 0·19 (2) | 0·40 (10) | 0·39 (0·07–2·15) |
| Pulmonary embolism | | | |
| Dose 1 | 1·02 (12) | 1·17 (23) | 1·17 (0·57–2·42) |
| Dose 2 | 1·03 (13) | 1·17 (23) | 0·91 (0·44–1·89) |
| Dose 3 | 1·48 (16) | 1·07 (27) | 1·07 (0·54–2·10) |
| Disseminated intravascular coagulation | | | |
| Dose 1 | 0·08 (2) | 0·04 (1) | n/a |
| Dose 2 | 0 | 0·04 (1) | n/a |
| Dose 3 | 0 | 0·04 (1) | n/a |
| Venous thromboembolism | | | |
| Dose 1 | 3·66 (43) | 3·11 (61) | 0·95 (0·62–1·43) |
| Dose 2 | 1·91 (24) | 3·11 (61) | 0·56 (0·34–0·92) |
| Dose 3 | 3·52 (38) | 3·42 (86) | 0·71 (0·47–1·08) |

*CI* confidence interval, *pdays* person-days, *n/a* not-applicable.

vaccines[12,13,15,23,24], we investigated additional haematological outcomes in this study (see Table S1).

Analyses of young and older age groups, or all ages combined, did not detect a significant association between myocarditis/pericarditis and the BBIBP-CorV and mRNA-based BNT162b2 vaccines. Previous case reports and studies of confirmed cases among individuals aged 12–39 years yielded an elevated IRR estimate of myocarditis/pericarditis within the first week after BNT162b2 vaccination, especially after the second dose[21,22]. In our study, very few cases of myocarditis/pericarditis (overall) were identified. Myocarditis is a form of cardiac inflammation underdiagnosed in practice[25], with clinical bias being directed towards myocardial ischemia or infarction. Thus, our use of diagnostic codes for myocarditis from routinely collected data suggest that ascertainment of cardiac inflammation after COVID-19 vaccination is likely to be under-represented. Our subgroup analyses also revealed that among individuals with previous occurrences CVDH events, there was no increased risk of the primary outcome events in the 1–21 days following dose 1, dose 2, and dose 3 of either vaccine. This agrees with the findings of a study conducted in Hong Kong, which also found no association between major adverse cardiovascular events following the administration of the BNT162b2 vaccine[26].

The other common events reported in this study, consistent with other studies[27], were stroke and venous thromboembolic events.

There have been reports showing associations between thromboembolic events and other COVID-19 vaccines[28], in particular the adenoviral-vectored vaccines, ChAdOx1 (Oxford-AstraZeneca)[12,13] and Ad26.COV 2.S (Janssen)[15,24]. For instance, a cohort study conducted across Denmark and Norway revealed that recipients of the ChAdOx1 vaccine experienced a higher incidence of venous thromboembolic events, including cerebral venous thrombosis, compared to the general population; the study observed an elevated number of these events in the vaccinated group, with the standardized morbidity ratio notably exceeding one[23]. In another study conducted in France, which included subjects younger than 75 years, the first dose of ChAdOx1 was found to be associated with myocardial infarction and pulmonary embolism in the second week of vaccination; in addition, the study found an increase in the number of hospitalizations associated with myocardial infarction in the second week after a single dose of the Ad26.COV 2.S vaccine[29]. Other systemic and local side effects after BNT162b2 and ChAdOx1 nCoV-19 vaccination have been self-reported by individuals at low frequencies[30].

In our assessment of the cardiovascular safety profile of COVID-19 vaccines, we found that the mRNA-based BNT162b2 and the inactivated BBIBP-CorV vaccines generally exhibit reassuring short-term safety with respect to CVDH events. Our findings show no significant association with cardiovascular diseases in individuals receiving the BBIBP-CorV vaccine. This stands in contrast to other vaccine platforms, notably the viral vector-based ChAdOx1 vaccine, which displayed in other studies a moderate association with pulmonary embolism and acute myocardial infarction[12,13,29]. Similarly, studies have suggested a potential risk for myocardial infarction associated with Ad26.COV 2.S vaccine[15,24,29], though further studies are necessary to substantiate these findings. It is important to note the scarcity of literature concerning the BBIBP-CorV vaccine's cardiovascular implications. A single case report has described a transient heart block that occurred post-vaccination in an 80-year-old patient with a pre-existing heart condition[31]. This paucity of data highlights the need for ongoing surveillance and research to fully elucidate the cardiovascular safety profile of the BBIBP-CorV vaccine and to ensure comprehensive post-vaccination monitoring across diverse populations. Overall, there has been no strong evidence that CVDH outcomes are associated with the two vaccines included in this study (including during randomized clinical trial studies)[32,33], but close monitoring is recommended to continue for CVDH outcomes after vaccination with all COVID-19 vaccines[34].

This study has several strengths. First, the UAE, namely Abu Dhabi, offered an ideal place to carry out this investigation given that at least two vaccines have been rolled out at speed and scale. Second, this was a large population-based study of data recorded retrospectively, which provided power to investigate the occurrence of these rare clinical events, which could not be assessed through clinical trials. Third, the ethnic diversity and demographic characteristics of the resident population provide an opportunity to evaluate the safety profile of these vaccines in a diverse population. Fourth, using a modified version of a rapid cycle analysis study design, the outcome incidence among vaccinees in a risk interval was compared with the outcome incidence among similar vaccinees who were in their comparison interval on the same calendar date. Thus, the comparison group for each outcome event was similar in demographic characteristics to the case and was in follow-up on the same day at the same site. This minimized the biases that can arise from variations in healthcare use during the pandemic (or at least, during the study period), as well as day-to-day variations. We also conducted several subgroup analyses to evaluate the risk of the outcomes within certain groups of interest and evaluate the robustness of our results. In addition, we note that the methodology employed in this study shares some characteristics with self-controlled designs, but it is not strictly a self-controlled case series (SCCS) study. A self-controlled approach, while advantageous in certain respects, presented limitations in our specific context. In a self-

controlled design, outcome events in the comparison interval invariably occur on later (or earlier) calendar dates than those in the risk interval. This temporal displacement could introduce bias, particularly in a rapidly evolving situation like a vaccine rollout, where external factors can significantly influence both exposure and outcome measures within a short time frame (e.g., daily variation in vaccine administration and occurrence of outcome events; see Fig. S1). Moreover, the chosen approach allowed for a more nuanced adjustment for time-invariant confounders. While SCCS inherently controls for time-invariant individual characteristics, it may not adequately adjust for time-varying confounders, particularly those related to external and systemic factors that can fluctuate substantially in rapidly evolving situations like a vaccine rollout[34–36]. Finally, this study design can be easily implemented and extended for monitoring the association of COVID-19 vaccines and other outcome events of interest[34].

There are some limitations that should be acknowledged. Firstly, for some of the outcomes of interest, the number of observed events in our study population was very low, which limited the statistical power of our analyses for our secondary outcomes. Secondly, the definition of the outcomes of interest relied on hospital admission diagnostic codes. Hence, cases were not confirmed using other data records such as clinical procedures or notes, and no medical audit was conducted. Thirdly, the risk may have been underestimated or missed if the real risk interval (post-vaccination) was longer than that considered in this study; in addition, only medically attended outcomes were included and certain individuals may have been lost to follow-up post-vaccination (who no longer interacted with the health system in the country); thus, analyses could have underestimated the risk if healthcare was not sought. Our sensitivity analysis restricted to those individuals with historical records attempts to address this but does not discount the possibility entirely. In addition, given the poorly documented information about the ethnicity of individuals, we relied instead on a "proxy" based on region of origin as defined by WHO. The small number of events in each regional group further constrained our ability to conduct a detailed sub-analysis by nationality or ethnicity, a factor that limits the findings of this study. Finally, in this study, we performed several comparisons, which may lead to some erroneous inferences, and although we used an established methodology for evaluating vaccine safety, we cannot determine whether our findings are causal. Consequently, careful interpretation is needed, especially for the borderline associations found.

In conclusion, this real-world population surveillance study of inactivated BBIBP-CorV and mRNA-based BNT162b2 vaccines can be used to complement other vaccine safety monitoring studies and systems running across the country. In general, no safety signal was detected for potential association between BBIBP-CorV and BNT162b2 vaccines and the selected cardiovascular and haematological events of interest in our study population.

## Methods

### Ethics approval
This study was approved by the Abu Dhabi Health Research and Technology Ethics Committee (approval number DOH/CVDC/2022/1466) of the Abu Dhabi Department of Health (s).

### Data source
We used de-identified data from the Abu Dhabi Health Information Exchange (HIE), known as "Malaffi", to identify COVID-19 vaccine exposure. The Malaffi HIE receives data from all licensed facilities within the Emirate of Abu Dhabi, and to date, it contains historical records from over 8 million individuals. This includes demographic information (including nationality), diagnosis codes, medications, immunization, laboratory results, in-hospital mortality information, and other procedures associated with patients' visits to the care facilities. Most notably, these data are linked to vaccination data

(including vaccine type, date, dose number, and site of administration), and the severe acute respiratory syndrome coronavirus 2 (SARS-CoV-2) infection data for all people in Abu Dhabi[6,37].

### Study design
A vaccine surveillance methodology, akin to a modified version of the rapid cycle analysis, was used; this design, originally developed for monitoring vaccine safety and reported in multiple studies, has been adapted for a single, comprehensive evaluation[16–19,35,36]. The analyses are well-adjusted for calendar date, which avoids biases that arise from day-to-day variation in health services. In addition, they are not confounded by time-stable characteristics during the study period, such as gender, ethnicity, and chronic conditions, which are controlled for.

### Study period and population
We examined potential safety signals of associations between the BBIBP-CorV and BNT162b2 vaccines and selected CVDH conditions during the COVID-19 vaccination program in the UAE, which commenced in December 2020. Due to data availability, people were considered eligible for inclusion in this study if they had received at least one vaccine dose, were at least 12 years old (at the time of vaccination), and were admitted to hospital or had an ER visit with the outcome of interest (i.e., follow-up study period) between 1 June 2021 and 30 June 2022. We note that data from the HIE covered the entirety of the study period.

### Outcomes
The outcomes in this study are selected CVDH conditions occurring post-administration of COVID-19 vaccine. These included acute myocardial infarction (AMI), non-haemorrhagic stroke (NHS), haemorrhagic stroke (HS), myocarditis or pericarditis (M/P), disseminated intravascular coagulation (DIC), pulmonary embolism (PE), and venous thromboembolism (VTE). The outcomes were identified as hospital admissions (inpatients) or ER visits due to an event of interest recorded within the study period. We used the *International Classification of Diseases and Related Health Problems, 10th Revision* codes to identify events of interest. Table S1 in supplemental information shows the codes corresponding to each outcome of interest.

We defined the primary outcome of this study as a composite event represented by the first hospitalization or ER visit in each day for any of the selected adverse events during the study period. Secondary outcomes of the study were outcome-specific for each one of the seven CVDH adverse events mentioned above.

### Exposures
The exposure variables were a first, second, or third dose of the BBIBP-CorV and BNT162b2 vaccines. We compared outcome incidence during a risk interval of days 1–21 after vaccination with the outcome incidence in vaccinated concurrent comparators. These comparators were vaccinees who were concurrently (i.e., on the same calendar day) in a comparison interval (i.e., 22–42 days after their most recent COVID-19 vaccination). For example, on 1 September individuals who were in their risk interval (or vaccinated between 11 and 31 August) were compared with individuals who, on 1 September, had had their most recent dose 22–42 days earlier (or last vaccinated between 21 July and 10 August). Note that vaccinated individuals contribute to the primary analyses as *exposed* when in a 21-day risk interval after dose 1, 2 or 3; they contributed as *non-exposed* when in the comparison interval 22–42 days after their most recent dose. A similar comparison interval has been used in other vaccine safety studies[34,36]. Note that for the comparison risk interval 22–42 days post the vaccine, individuals administered any additional COVID-19 vaccine dose within the period 1–21 days were excluded from the comparison group.

To avoid the confounding effect of SARS-CoV-2 infection on the outcomes of interest, individuals who had a SARS-CoV-2 positive test

recorded in the period preceding or during the risk interval or the concurrent comparator interval were censored for up to 30 days following the positive test[20]. Similarly, individuals with recorded in-hospital death were censored at the date of death. We note that the death certificate was unavailable at the time of this study.

In addition, to reduce the possibility of confounding by demographic factors and factors associated with calendar time, we conducted analyses within strata defined by 5-year age group, gender (male, female or unknown), *nationality*, vaccine administration site, and calendar week. For *nationality*, individuals were categorized into one of the following (according to their nationality): UAE Nationals, GCC Members (countries that are members of the Gulf Cooperation Council), and the six World Health Organization (WHO) regions (those missing nationality were categorized as unknown)[38].

We also categorized individuals into high and low-risk groups identified from a subset of the study population having at least one medical encounters (inpatient, outpatient, or ER visit) in the 12-month lookback period preceding the first recorded exposure during the study period. Diagnostic codes for the outcomes of interest were used to identify the pre-existence of one or more outcome events before vaccination. Those with previous history of the selected CVDH conditions were assigned as high-risk and those with no previous history of the selected conditions to low-risk (those with no encounters recorded in the 12-month lookback period were categorized as unknown).

### Statistical analysis

We described the characteristics of the study population in terms of age, gender, nationality, and risk groups (per vaccine type and dose number). Poisson regression models were used to estimate an adjusted incidence rate ratio (IRR) and corresponding 95% confidence interval (CI), estimating the relative rate of hospital admissions or ER visits related to the outcomes of interest in exposure risk periods relative to the comparison periods. Separate analyses were carried out for the primary outcome, and each (secondary) outcome of interest. The models were fitted with an offset for the person-days in each risk or comparison interval, adjusting for age group, gender, nationality, administration site and week (including the exposure term).

To investigate possible age or gender-dependent association between vaccine exposure and outcome, we conducted subgroup analyses amongst those aged under 40 years and those aged 40 years and older, and by gender. We also conducted analyses restricted to those either in the high or low-risk groups.

To compare the incidence rates during the risk period itself, we included analyses of three risk intervals by comparing the outcome incidence during a risk interval of 1–7, 8–14, and 15–21 days after vaccination.

In addition, to evaluate our study's statistical power, we conducted a post-hoc power analysis. The analysis involved determining the sample size required to detect specified effect sizes of interest (e.g., a 20% or 30% increase in risk) with desired power of 80% at a significance level ($\alpha$) of 0.05. This analysis was performed using a Poisson regression, considering the characteristics of the study population and outcome variables for the estimation of incidence rates of the primary (composite) outcome and assuming groups of equal size.

### Reporting summary

Further information on research design is available in the Nature Portfolio Reporting Summary linked to this article.

## Data availability

According to Abu Dhabi Department of Health regulations, individual-level data cannot be shared openly. Specific requests for remote access to de-identified data should be referred directly to IRB, DOH Research Committee, DOH Medical Research and Development Division (medical.research@doh.gov.ae).

## Code availability

The code used for this study can be made available upon specific request to the corresponding author(s).

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

## Acknowledgements
This study is conducted in collaboration with the Department of Health Abu Dhabi which granted researchers remote and secure data access to de-identified datasets. Views and opinions expressed in the text are those of the authors and do not necessarily reflect the official policy or position of the Department of Health or other affiliated institutions.

## Author contributions
M.A.F.P. contributed to study conceptualisation, data curation, formal analysis, investigation, methodology, validation, visualisation, writing – original draft, and writing – review & editing. M.S. contributed to study conceptualisation, formal analysis, investigation, methodology, project administration, resources, validation, visualisation, writing – original draft, and writing – review & editing. M.A. contributed to study conceptualisation, data curation, supervision, validation, visualisation, and writing – review & editing. Y.N. contributed to the study conceptualisation, methodology, validation, visualisation, and writing – review & editing. S.K. contributed to conceptualisation, methodology, project administration, resources, and writing – review & editing. M.A.F.P., M.A., and Y.N. had complete access to the data throughout the study period and have accessed and verified the data.

## Competing interests
M.A.F.P., M.S. and S.K. were employed by G42 Healthcare at the time of study and are currently employed by M42 Health. M.A. and Y.N. are employed by the Department of Health, Abu Dhabi at the time of study. The authors declare that they have no known competing financial interests or personal relationships that could have appeared to influence the works reported in this paper. There was no external funding source for this study.
