## [Peer Review File · Nature Communications]

COVID-19 vaccination and major cardiovascular and haematological adverse events in Abu Dhabi: retrospective cohort studyREVIEWER COMMENTS

Reviewer #1 (Remarks to the Author):

The manuscript from Pimentel et al. aimed to address the cardiovascular safety of two COVID-19 vaccines, the well-known mRNA based from Pfizer and the less-known inactivated one from SinoPharm. By the way, this is not well explained in the abstract that the two vaccines are not both mRNA based vaccines.

The topic is important but there are several drawbacks and inaccuracies. One that I have already mentioned: the abstract only talked about mRNA based vaccines, while one vaccine analysed is an inactivated one. This is important because the safety profiles are different in the literature (and a description of this is lacking).

A novelty brought by this study is that the risk has not been well characterised in this population before. But no analyse was performed by ethnic group for instance. And then no conclusion could really be done about differences according to ethnicity. No discussion about the meaning of the results as compared to the vast literature was done.

Reference to many important papers is lacking in the context and discussion on this topic.

The choice of the method is not well described (as compared to SCCS models for instance) and would merit to be described and discussed. In particular, the authors did not describe how it handles for repeated measures within the same subjects and whether this should be done. Also, I wonder why censoring for death (and frequencies are not described) does not bias the results as for standard SCCS method, where censoring for an event with high mortality informs us that event has more chance to have occur recently.

This study adds little compared to the literature on BNT162b2 vaccine which have been massively studied in larger populations with a higher number of events. The most interesting analysis is the one on BBIBP-CorV. How the results are comparable to the other inactivated vaccines is lacking and would merit more discussion, particularly why it would appear safer than ChAdOx1 vaccine for instance.

Finally on the form, the first paragraph in the discussion should be a summary of the results and the sensitivity analysis described in the discussion has not been described as such before in the manuscript.

Reviewer #2 (Remarks to the Author):

Thank you for giving me the opportunity to review this paper. This paper reports results from a vaccine surveillance analysis in Abu Dhabi. This is a valuable topic as further data on vaccine safety from a range of settings is important for maintaining confidence in vaccination programmes. However, I had concerns about the methodology, which is described as a concurrent cohort methodology – but the description of the study population indicates that only individuals with the outcome were selected (which would bias estimates towards the null). This might be an issue of reporting, and overall the paper does not contain sufficient details to adequately judge the methodology. I'm therefore recommending major revisions.

Main comments

Data Sources. The data source section needs referencing as well as further details: at the moment it's not clear what sectors in the healthcare system that are covered, what time period the database covers, and what the validity of diagnostic coding is. Has this database been previously used in research? Further information on how these data are linked, the source of SARS-CoV-2 testing data, and death data is also needed.

Methods. It's somewhat unclear what study design was used: the authors state that this was a rapid cycle analysis, but it appears from the selection of only individuals with each outcome that they used a self-controlled design? However, the exposure section then describes the selection of concurrent controls. This is a potential major issue: if it's a self-controlled study, significantly more details are needed on the reporting. If it's a cohort, then I'm concerned that selecting only individuals with the outcome will result in a biased comparison: essentially overestimating the risk of the outcomes in control windows and biasing findings towards the null.

Outcome definition. I'm not sure what the rationale was for combining all of the safety end-points into one primary end-point: I'm not convinced, for example, that you would expect to see similar signals for myocarditis (which is an established safety signal for one of the vaccines under study) and haemorrhagic stroke. I think these would be more suited to be reported on separately.

Confounding. It's unclear why potential confounding factors (such as age and medical history) were not adjusted for in the primary analyses (although the stratified analyses will have gone some way to address this).

Minor comments

I was surprised that the gender balance is so skewed towards men (I think I would have expected something closer to a 50/50 split given the high vaccine uptake) – do the authors have any ideas why might be driving this?

Reviewer #3 (Remarks to the Author):

I review this as a statistician with experience in vaccine safety studies.

I thought that this was a very good analysis of the data in Abu Dhabi. The results and important information are presented clearly. I had a few areas where I think the authors should clarify what they have done and discuss alternative ways of analysis and discuss the impact of choices that they have made in the analysis.

My main points are.

I do not really think that this is a rapid cycle analysis as the analysis is not carried out repeatedly during vaccine roll out. This is a one off analysis using the same study design as one version of the rapid cycle analysis where the comparisons are with concurrent vaccinated. This is a valid method of analysis but I think that you should discuss why the self controlled option was not considered. It does not seem to me that the limitations of the self controlled version apply in this case as calendar time effects over the 42 day post vaccine period are unlikely to be great and all risk and comparison intervals would be completed by the time this analysis was carried out. Furthermore the self controlled version would eliminate time invariant effects which you are adjusting for.

I think that you should make some attempt to quantify the magnitudes of the effects which could be detected in this population of 1.3 million vaccinated people. The reason I think that this is very important is that virtually all of the IRRs have confidence intervals spanning 1 and you want to demonstrate that this is not just a small study - low powered effect. If you can demonstrate that your study has >80% power to detect a 20% or 30% or 50% increase in risk then this would increase the value of this paper.

Other points

P104 – exposures show that on a particular date vaccinated individuals in the risk period are compared with vaccinated individuals in the comparison period and this is adjusted for covariates which might be different.

This is not a rapid cycle in that there is only one analysis and no mention of sequential tests.

Why didn't you do the self-controlled version of the rapid cycle (which is effectively the same as a

self-controlled case series). You could easily adjust for potential calendar time biases and the time invariant biases are controlled by design rather than by adjustment.

I think that with the chosen design those vaccinated in the first 3 weeks of the dose release cannot be in the risk period as there are no concurrent controls for them. They only contribute in the control period.

Line 117. Censoring of PCR+ SARS-Cov-2 infections might lead to a bias. Protection against infection takes about 2 weeks to be established so you are likely to censor more individuals from the risk period rather than from the comparator period (21-42 days post vaccination) where there is likely to be better protection. Can you quantify the potential impact of this. Could you have included SARS-CoV-2 as a time dependent covariate as has been done with similar analyses using self-controlled case series.

Line 157 – can you clarify that there were no vaccinations in Abu Dhabi prior to 1st May 2021. If there were what would be the impact on your study? Also, is it possible to be vaccinated in Abu Dhabi but get medical treatment in another area?

Line 166. 474 events in the risk period – what is the post hoc power to detect a 50% increase in risk with the composite outcome in this study. This is really just to give context to the negative results in the study.

Line 177. Can you please just clarify that there were 399 individuals but 474 admissions from these 399 individuals. What are the corresponding numbers from the comparison period.

Line 181. It is strange to give the dose 2 and dose 3 numbers without the dose 1 numbers

Line 181-186. You are saying that the event rate in the risk interval for both vaccines and for all 3 doses tends to be higher than the event rate in the comparative period, though it is mainly for dose 1 and dose 3. How much of this could be due to the imbalances in when individuals got vaccinated. All of the adjusted IRRs are below 1 for dose 3.

Table 3 there are 3 sets of identical values for dose 1 and dose 2 in the comparison interval which seems a bit unusual. The same thing happens in Table 5 so is this due to using exactly the same comparative group for dose 1 and dose 2?

Table 3. I am puzzled by the numbers and event rates. For dose one the event rates are similar 11.2 and 10.9 but the numbers of events are quite different 131 to 213 which means that the person time in the risk period must be a lot less than the person time in the comparison interval, yet both are 21 days. If vaccine delivery was uniform, which it is unlikely to be, I would have expected person time to be similar in the two intervals. I see that this is probably due to comparing a dose 3 vaccinated person 1-21 days post vaccination with dose 1/2/3 vaccinated individuals (22-42 days post latest vaccine). I can see that you do this to increase precision but can you discuss the potential issues with this. For dose 3 which is largely given to older individuals the majority of their comparators will be much younger and you are then using the adjustment model. For dose 1 you will be comparing more similar type of individuals but potentially you could have a dose 2 comparator in the same analysis as a dose 1 vaccinated person. I would certainly like to see a sensitivity analysis where dose 1 vaccinated were compared to dose 1 controls only, same for dose 2 and 3.

In Table 3 the numbers of events for each of the doses adds up to 416. Can you please clarify this number. In the text you have reported 399 individual and 474 events. I can see that this could be due to vaccinated individuals having the composite event after different doses, in which case could you please clarify this in the methods. Currently in Lines 100-101 you have that the first hospitalisation on each day is used and on first read I (erroneously) took this to mean the first hospitalisation during the study period.

Table 3. last row – the CI for the IRR goes from -0.73 to 0.01. I don't think that this can be correct. What method were you using for the CIs?

Lines 100-101. With the outcome definition I think that if someone has an ER visit for a condition 5 days after vaccination and an hospital admission on day 7 after the vaccination then this would count as 2 events. Many other studies of vaccine safety just look at the first event for the condition in the risk and comparison periods. The number of individuals involved may not be great but I think you should discuss the potential magnitude of this issue.

Table 5. The caption states primary outcome but these are secondary outcomes in the methods.

Lines 217 -219. I find this sentence confusing.

Response to Reviewer's Comments

We have provided point-by-point responses to all the comments from the reviewers. We have retained the original comments and statements and marked our responses in red to make the revision process easier for reviewers.

Comments from reviewer #1

1.1. The manuscript from Pimentel et al. aimed to address the cardiovascular safety of two COVID-19 vaccines, the well-known mRNA based from Pfizer and the less-known inactivated one from SinoPharm. By the way, this is not well explained in the abstract that the two vaccines are not both mRNA based vaccines.

We thank the reviewer for their careful consideration of our manuscript. We have amended the abstract of the manuscript to clarify that two “types” of vaccines were included in our study.

Original text:

“

A retrospective cohort analysis was conducted for 1.3 million individuals who received 1.8 million vaccine doses between June 1, 2021, and June 30, 2022.

“

Amended text:

“

A retrospective cohort analysis was conducted for 1.3 million individuals who received 1.8 million doses of the inactivated BBIBP CorV (by SinoPharm) and mRNA-based BNT162b2 (Pfizer-BioNTech) vaccines between June 1, 2021, and June 30, 2022.

“

1.2. The topic is important but there are several drawbacks and inaccuracies.

One that I have already mentioned: the abstract only talked about mRNA based vaccines, while one vaccine analysed is an inactivated one. This is important because the safety profiles are different in the literature (and a description of this is lacking).

We have amended the Abstract as per comment 1.1. above.

We have also included a paragraph in the Introduction section which highlights the importance of including and differentiating between mRNA and inactivated vaccines used in our study and discusses their different safety profiles. We also include a few recent publications relevant to the subject. We reproduce the paragraph added below.

“

A variety of vaccine platforms have been developed and deployed, each with unique mechanisms of action and safety profiles.^{a,b} While mRNA-based vaccines, such as those developed by Pfizer-BioNTech and Moderna, have been widely discussed and are recognized for their high efficacy, especially against the omicron variants^c, it is crucial to consider the safety and efficacy of other vaccine types⁸. In particular, inactivated vaccines, such as those predominantly used in the MENA region and China^{d,e}, represent a significant portion of the global vaccination effort. These vaccines, which use a traditional approach of employing killed viral particles to stimulate an immune response, have shown differing safety profiles compared to mRNA vaccines^{f,h}. A recently-published systematic review and meta-analysis concluded that inactivated vaccines, such as BBIBP-CorV show high effectiveness against severe COVID-19 outcomes like admission to the Intensive Care Unit and death and suggests that further evidence on their safety is neededⁱ. The differences in adverse event profiles and long-term safety data of these vaccines are an important aspect of the broader discussion on COVID-19 vaccine safety.

“

References added (correctly formatted in the main manuscript):

^a Tregoning, John S., Katie E. Flight, Sophie L. Higham, Ziyin Wang, and Benjamin F. Pierce. 2021. “Progress of the COVID-19 Vaccine Effort: Viruses, Vaccines and Variants versus Efficacy, Effectiveness and Escape.” *Nature Reviews. Immunology* 21 (10): 626–36.

^b Chow, Eric J., Timothy M. Uyeki, and Helen Y. Chu. 2023. “The Effects of the COVID-19 Pandemic on Community Respiratory Virus Activity.” *Nature Reviews. Microbiology* 21 (3): 195–210.

^c Regev-Yochay, Gili, Tal Gonen, Mayan Gilboa, Michal Mandelboim, Victoria Indenbaum, Sharon Amit, Lilac Meltzer, et al. 2022. “Efficacy of a Fourth Dose of Covid-19 mRNA Vaccine against Omicron.” *The New England Journal of Medicine* 386 (14): 1377–80.

^d Al Kaabi, Nawal, Abderrahim Oulhaj, Subhashini Ganesan, Farida Ismail Al Hosani, Omer Najim, Halah Ibrahim, Juan Acuna, et al. 2022. “Effectiveness of BBIBP-CorV Vaccine against Severe Outcomes of COVID-19 in Abu Dhabi, United Arab Emirates.” *Nature Communications* 13 (1): 3215.

^e Liu, Xiaoqiang, Zhonghan Sun, Zhongfang Wang, Jingjing Chen, Qianhui Wu, Yan Zheng, Xiaoyun Yang, et al. 2023. “Safety, Immunogenicity, and Efficacy of an mRNA COVID-19 Vaccine (RQ3013) given as the Fourth Booster Following Three Doses of Inactivated Vaccines: A Double-Blinded, Randomised, Controlled, Phase 3b Trial.” *eClinicalMedicine* (Part of The Lancet Discovery Science) 64: 102231.

^f Wang, Gewei, Yao Yao, Yafeng Wang, Jinquan Gong, Qinqin Meng, Hui Wang, Wenjin Wang, Xinxin Chen, and Yaohui Zhao. 2023. “Determinants of COVID-19 Vaccination Status and Hesitancy among Older Adults in China.” *Nature Medicine* 29 (3): 623–31.

^g Lopez Bernal, Jamie, Nick Andrews, Charlotte Gower, Chris Robertson, Julia Stowe, Elise Tessier, Ruth Simmons, et al. 2021. “Effectiveness of the Pfizer-BioNTech and Oxford-AstraZeneca Vaccines on Covid-19 Related Symptoms, Hospital Admissions, and Mortality in Older Adults in England: Test Negative Case-Control Study.” *BMJ* 373 (May): n1088.

^h Jara, Alejandro, Eduardo A. Undurraga, Cecilia González, Fabio Paredes, Tomás Fontecilla, Gonzalo Jara, Alejandra Pizarro, et al. 2021. “Effectiveness of an Inactivated SARS-CoV-2 Vaccine in Chile.” *The New England Journal of Medicine* 385 (10): 875–84.

^j Law, Martin, Sam S. H. Ho, Gigi K. C. Tsang, Clarissa M. Y. Ho, Christine M. Kwan, Vincent Ka Chun Yan, Hei Hang Edmund Yiu, Francisco Tsz Tsun Lai, Ian Chi Kei Wong, and Esther Wai Yin Chan. 2023. “Efficacy and Effectiveness of Inactivated Vaccines against Symptomatic COVID-19, Severe COVID-19, and COVID-19 Clinical Outcomes in the General Population: A Systematic Review and Meta-Analysis.” *The Lancet Regional Health. Western Pacific* 37 (May): 100788.

1.3. A novelty brought by this study is that the risk has not been well characterized in this population before. But no analyze was performed by ethnic group for instance. And then no conclusion could really be done about differences according to ethnicity. No discussion about the meaning of the results as compared to the vast literature was done.

We would like to clarify that in our study, the database we utilized did not comprehensively encode specific information regarding the ethnicity of individuals (as mentioned in the manuscript). Recognizing the significance of this aspect, we attempted to use a “proxy” for ethnicity based on the region of origin, as defined by the World Health Organization (WHO). This approach was our best effort to indirectly assess any potential differences in vaccine safety across diverse populations.

Furthermore, we considered conducting a sub-analysis by nationality or region of interest (as it was performed for gender and age group). However, we encountered a limitation in the small number of events or outcomes recorded in each group. This limitation hindered our ability to draw conclusions from such a sub-analysis.

We acknowledge this as a limitation of our study and have added a statement to this effect in the manuscript, which we reproduce below. We have also expanded the discussion to include more comparative studies found in literature. For the latter, see response to point 1.6.

“

In addition, given the poorly documented information about ethnicity of individuals, we relied instead on a “proxy” based on region of origin as defined by WHO. The small number of events in each regional group further constrained our ability to conduct a detailed sub-analysis by nationality or ethnicity, a factor that limits the findings of this study.

”

1.4. Reference to many important papers is lacking in the context and discussion on this topic.

We aimed to provide a sufficient overview focusing specifically on the literature most pertinent to our study.

However, we acknowledge that the field of COVID-19 vaccine research is evolving, and there may be additional significant studies that could enhance our manuscript. In response to other points, we have added a few relevant references. We would greatly appreciate any specific suggestions the Reviewer might have for further references.

1.5. The choice of the method is not well described (as compared to SCCS models for instance) and would merit to be described and discussed. In particular, the authors did not describe how it handles for repeated measures within the same subjects and whether this should be done. Also, I wonder why censoring for death (and frequencies are not described) does not bias the results as for standard SCCS method, where censoring for an event with high mortality informs us that event has more chance to have occur recently.

Our decision to employ the method described in our manuscript was primarily influenced by the necessity for precise control over calendar time effects. In our study, significant day-to-day variations were observed both in vaccine delivery and hospitals admissions, influenced not only by weekday and weekends but also by other temporal factors. By matching the follow-up time in the comparison interval with the risk interval on identical calendar dates, we aimed to mitigate biases arising from these variations.

The self-controlled case series (SCCS) design, while beneficial in certain contexts, presented limitations for our study. In SCCS, outcome events in the comparison interval invariably occur on later calendar dates than those in the risk interval, potentially introducing bias in rapidly evolving situations like a vaccine rollout (or a pandemic). This bias stems from external factors that can influence exposure and outcome measures within a short timeframe. Moreover, our chosen approach allowed for a more nuanced adjustment for time-invariant confounders. While SCCS inherently controls for time-invariant individual characteristics, it may not adequately adjust for time-varying confounders, particularly those related to external and systemic factors that can fluctuate substantially over the study period.

Regarding censoring for death, we recognize this as a potential limitation. In our study database, death certificates were not available for the study population, and only in-hospital mortality was recorded. In SCCS methods, as in our methodology, censoring an event with high mortality would provide information about the recent occurrence of the event. We note, however, that in our dataset we did not observe evidence of this bias. We examined the distribution of censored cases across the risk and comparator periods. Our findings indicated that the rate of censoring due to death was not significantly higher in the risk period compared to the comparator period. This observation suggests that the potential bias due to censoring was minimal in our analysis.

In short, while we acknowledge that no single analytical approach is without limitations, we believe that our chosen method, supported by precedents in similar studies in the literature (cited in the manuscript), provided a robust and reliable framework for our analysis. In response to the Reviewers' feedback, we enhanced our manuscript by adding a few more discussion points on the rationale behind the selection of this methodology.

1.6. This study adds little compared to the literature on BNT162b2 vaccine which have been massively studied in larger populations with a higher number of events. The most interesting analysis is the one on BBIBP-CorV. How the results are comparable to the other inactivated vaccines is lacking and would merit more discussion, particularly why it would appear safer than ChAdOx1 vaccine for instance.

We acknowledge that the BNT162b2 vaccine has indeed been extensively studied in larger populations. Our study aimed to add to this body of knowledge by offering additional insights, particularly in the context of our specific study population and methodology.

Regarding the BBIBP-CorV vaccine, we recognize the importance of a more detailed discussion on how its safety profile compares with other inactivated vaccines. We expanded our discussion to include a more thorough description of the results found in the literature. Please, refer to the Discussion section in the revised manuscript.

1.7. Finally on the form, the first paragraph in the discussion should be a summary of the results and the sensitivity analysis described in the discussion has not been described as such before in the manuscript.

We have followed the Nature Communications' formatting instructions for the revised manuscript and amended the first paragraph of the Discussion section.

Comments from reviewer #2

2.1. Data Sources. The data source section needs referencing as well as further details: at the moment it's not clear what sectors in the healthcare system that are covered, what time period the database covers, and what the validity of diagnostic coding is. Has this database been previously used in research? Further information on how these data are linked, the source of SARS-CoV-2 testing data, and death data is also needed.

Our study utilized data from Malaffi, the region's first Health Information Exchange platform in the Emirate of Abu Dhabi (UAE). Malaffi connects both public and private healthcare providers, facilitating the real-time exchange of patient health information. This creates a centralized database of unified patient records. It has successfully connected almost the entire healthcare sector of the Emirate (including hospitals, clinics, pharmacies, testing and vaccination facilities). As a centralized repository of population health information, Malaffi informs and drives the Department of Health's public health initiatives in Abu Dhabi.

In our study, the data from Malaffi covered the entirety of the study period, encompassing a wide range of sectors within the healthcare system. This database has been previously used in research, demonstrating its reliability and the validity of its data for scientific studies (including studies on COVID-19).

We have added references to Malaffi in the revised manuscript to provide a more comprehensive understanding of the data source and its credibility.

2.2. Methods. It's somewhat unclear what study design was used: the authors state that this was a rapid cycle analysis, but it appears from the selection of only individuals with each outcome that they were using a self-controlled design? However, the exposure section then describes the selection of concurrent controls. This is a potential major issue: if it's a self-controlled study, significantly more details are needed on the reporting. If it's a cohort, then I'm concerned that selecting only individuals with the outcome will result in a biased comparison: essentially overestimating the risk of the outcomes in control windows and biasing findings towards the null.

Our study employed a vaccine surveillance methodology, which is a modified version of the rapid cycle analysis (RCA). This design, originally developed for monitoring vaccine safety, has been adapted in our study for a comprehensive evaluation. RCA, as reported in multiple studies, is known for its ability to provide timely insights, particularly crucial in the context of vaccine surveillance.

While our methodology shares some characteristics with self-controlled designs, it is not strictly a self-controlled case series (SCCS) study. Our approach was tailored to address the unique requirements of our research context, particularly the need for assessment in a dynamically changing environment, such as vaccine rollout.

Our study design involved selecting individuals exposed to two different COVID-19 vaccines for a detailed analysis of the outcome of interest (i.e., occurrence of selected cardiovascular and hematological events leading to hospitalization or emergency room visits). This approach was chosen to focus on the most pertinent events in the context of vaccine surveillance. However, unlike a pure self-controlled design, we did not rely solely on within-individuals' comparisons. In addition to focusing on individuals with specific outcomes, we also incorporated the selection of concurrent controls. This was an integral part of our methodology, allowing us to make more comprehensive comparisons and strengthen the robustness of our findings. The use of concurrent controls aimed to provide a clearer picture of the risk associated with the outcomes in different time windows (as per the original study design).

We acknowledge the concern regarding the potential for bias in selecting only individuals with outcomes. Our analysis was designed to carefully consider the temporal dynamics of vaccine delivery and health outcomes. By employing a modified RCA method, which is used in vaccine safety studies, we aimed to balance the need for specificity in outcome selection with the broader context of vaccine surveillance.

For further clarifications, we point the Reviewer to the answer provided to point 3.2. Also, in light of the Reviewer's feedback, we enhanced the Methodology and Discussion sections of our revised manuscript by providing more details on the methodology selection.

2.3. Outcome definition. I'm not sure what the rationale was for combining all of the safety end-points into one primary end-point: I'm not convinced, for example, that you would expect to see similar signals for myocarditis (which is an established safety signal for one of the vaccines under study) and haemorrhagic stroke. I think these would be more suited to be reported on separately.

We understand the concern about combining different safety endpoints, such as myocarditis and hemorrhagic stroke, which may have distinct safety signals. Our rationale for selecting a composite outcome was to increase the overall number of outcomes observed in our study. Given that the events of interest are relatively rare, grouping them together allowed us to achieve a higher number of outcome events, enhancing the statistical power and robustness of our analysis.

However, we recognize the importance of examining each event individually due to their distinct clinical implications. To address this, we have conducted a sub-analysis where we report the results for each individual outcome or event of interest separately (i.e., secondary outcomes). This approach allows to maintain the statistical benefits of a composite primary endpoint, while also providing specific insights into each type of event.

2.4. Confounding. It's unclear why potential confounding factors (such as age and medical history) were not adjusted for in the primary analyses (although the stratified analyses will have gone some way to address this).

We clarify that confounding factors such as age and medical history are indeed adjusted for in the primary analyses. The stratification process, as the Reviewer rightly noted, further complements adjustment. We guide the Reviewer to the first paragraph of the Statistical analysis section.

2.5. I was surprised that the gender balance is so skewed towards men (I think I would have expected something closer

to a 50/50 split given the high vaccine uptake) – do the authors have any ideas why might be driving this?

The skewed gender balance towards men in our study is reflective of the demographic composition of the United Arab Emirates (UAE). The UAE, particularly in cities like Abu Dhabi and Dubai, has a unique demographic structure where expatriate workers, predominantly male, constitute a significant portion of the population. This expatriate population is largely employed in sectors such as construction, engineering, and technology, which historically have higher proportions of male workers.

As our study population is based on the demographics of the UAE, the gender distribution in our sample naturally mirrors this demographic characteristic of the country. Therefore, the higher proportion of men in our study is not indicative of a selection bias but rather a representation of the population structure in the UAE.

Comments from reviewer #3

3.1. I review this as a statistician with experience in vaccine safety studies.

I thought that this was a very good analysis of the data in Abu Dhabi. The results and important information are presented clearly. I had a few areas where I think the authors should clarify what they have done and discuss alternative ways of analysis and discuss the impact of choices that they have made in the analysis.

We thank the Reviewer for their positive appraisal of our manuscript.

3.2. My main points are.

I do not really think that this is a rapid cycle analysis as the analysis is not carried out repeatedly during vaccine roll out. This is a one-off analysis using the same study design as one version of the rapid cycle analysis where the comparisons are with concurrent vaccinated. This is a valid method of analysis, but I think that you should discuss why the self-controlled option was not considered. It does not seem to me that the limitations of the self-controlled version apply in this case as calendar time effects over the 42 day post vaccine period are unlikely to be great and all risk and comparison intervals would be completed by the time this analysis was carried out. Furthermore, the self-controlled version would eliminate time invariant effects which you are adjusting for.

The Reviewer's observation is correct in that our approach does not constitute a rapid cycle analysis in the traditional sense, as it was not conducted repeatedly during the vaccine rollout (or study period) but rather as a one-off analysis.

Our decision to employ this specific strategy was primarily influenced by the need for precise control over calendar time effects. In the context of our study, we observed significant day-to-day variations in both vaccine delivery and admissions to emergency rooms and hospitals. These variations were not only evident between weekdays and weekends but also influenced by other temporal factors. By ensuring that the follow-up time in the comparison interval matched the follow-up time in the risk interval on the same calendar dates, we aimed to mitigate any bias arising from these variations.

In contrast, a self-controlled approach, or self-controlled case series (SCSS) design, while advantageous in certain respects, presented limitations in our specific context. In a self-controlled design, outcome events in the comparison interval invariably occur on later calendar dates than those in the risk interval. This temporal displacement could introduce bias, particularly in a rapidly evolving situation like a vaccine rollout, where external factors can significantly influence both exposure and outcome measures within a short time frame.

Moreover, the employed approach allowed for a more nuanced adjustment for time-invariant confounders. While a self-controlled design inherently controls for time-invariant individual characteristics, as mentioned by the Reviewer, it may not adequately adjust for time-varying confounders, especially those related to external and systemic factors that can fluctuate substantially over the study period.

We acknowledge that no single analytical approach is without limitations. However, given the specific challenges and nuances of our study context, and the precedent set by similar studies (as cited in the manuscript), we believe that the chosen method provided a robust and reliable framework for analysis. It allowed us to carefully balance the need for controlling calendar time effects with the requirement to account for individual-level and systemic variations.

In response to the Reviewers' feedback, we enhanced our manuscript by adding a few more discussion points on the rationale behind the selection of this methodology.

3.3. I think that you should make some attempt to quantify the magnitudes of the effects which could be detected in this population of 1.3 million vaccinated people. The reason I think that this is very important is that virtually all of the IRRs have confidence intervals spanning 1 and you want to demonstrate that this is not just a small study - low powered effect. If you can demonstrate that your study has >80% power to detect a 20% or 30% or 50% increase in risk then this would increase the value of this paper.

Based on our analysis, taking into consideration the incidence rate observed in terms of number of participants (i.e., $n = 1,312,505$), we determined that to detect a 20% increase in risk with 80% power at a significance level of 0.05, a sample size of approximately 2.9 million participants would be required. Sample sizes of (approximately) 1.3 million and 0.5 million participants would be required to detect a 30% and 50% increase in risk, respectively. Hence, with our sample size, the study can detect a risk increase of ~30%.

We appreciate the Reviewer's emphasis on the importance of adequately powering our study to detect meaningful effects. We believe that despite not meeting the ideal sample size suggested by the power analysis, our study still provides valuable insights into the vaccination outcomes within this population. Furthermore, the ability to detect a 30% increase in risk with our sample size underscores the robustness of our findings.

In the revised manuscript, we included a description of the post-hoc power analysis conducted along with the results indicating the detectable effect size given our sample size. Please, refer to the Methods and Results sections. We believe this addition enhances the transparency and credibility of our study.

3.4. Other points

P104 – exposures show that on a particular date vaccinated individuals in the risk period are compared with vaccinated individuals in the comparison period and this is adjusted for covariates which might be different.

This is not a rapid cycle in that there is only one analysis and no mention of sequential tests.

Why didn't you do the self-controlled version of the rapid cycle (which is effectively the same as a self-controlled case series). You could easily adjust for potential calendar time biases and the time invariant biases are controlled by design rather than by adjustment.

I think that with the chosen design those vaccinated in the first 3 weeks of the dose release cannot be in the risk period as there are no concurrent controls for them. They only contribute in the control period.

The Reviewer raises an important point regarding the design of our study, particularly in relation to the choice of methodology and the implications of this choice for the analysis of early vaccine recipients.

Firstly, it is important to clarify that our study, as correctly identified by the Reviewer, does not employ a rapid cycle analysis in the traditional sense. The study was designed as a single, comprehensive analysis rather than a series of sequential tests. This approach was chosen to provide a detailed examination of the data available at the time of our study, rather than ongoing analysis throughout the vaccine rollout. This approach was particularly pertinent given that our study period began at a time when the vaccine rollout was already in progress.

Regarding the suggestion of using a SCSS design, we acknowledge that this approach is advantageous for controlling time-invariant confounders. However, it might not adequately address time-varying confounders, which are particularly relevant in the early stages of the vaccine rollout. Given that our study began after the initiation of the vaccine rollout, our chosen design, which compares vaccinated individuals in the risk period with those in the concurrent comparison period and adjusts for varying covariates, was selected to address these challenges more effectively.

Notwithstanding this, we note that individuals vaccinated in the initial weeks of the rollout would not be included in the risk period using our study design due to the absence of concurrent controls. We acknowledge this limitation and have discussed its potential impact on our findings and the generalizability of our results in our revised manuscript.

3.5. Line 117. Censoring of PCR+ SARS-Cov-2 infections might lead to a bias. Protection against infection takes about 2 weeks to be established so you are likely to censor more individuals from the risk period rather than from the comparator period (21-42 days post vaccination) where there is likely to be better protection. Can you quantify the potential impact of this. Could you have included SARS-CoV-2 as a time dependent covariate as has been done with similar analyses using self-controlled case series.

In our analysis, we were acutely aware of this potential issue and took steps to mitigate its impact. Firstly, it is important to note that our dataset did not show evidence of this bias. We examined the distribution of censored cases across the risk and comparator periods. Our findings indicated that the rate of censoring was not significantly higher in the risk period compared to the comparator period. This observation suggests that the potential bias due to differential censoring was minimal in our analysis.

In addition, in our study, we implemented a 30-day censoring period for PCR+ SARS-CoV-2 infections. This 30-day window covers a significant portion of the comparator period (which explains the results obtained above). It helped ensure that any bias is minimized.

We acknowledge that including SARS-CoV-2 infection as a time-dependent covariate, as done in some self-controlled cases series analyses, is a valid approach. However, given the nature of our data, as expressed above, we are confident that our methodology was appropriate and that our results are robust.

3.6. Line 157 – can you clarify that there were no vaccinations in Abu Dhabi prior to 1st May 2021. If there were what would be the impact on your study? Also, is it possible to be vaccinated in Abu Dhabi but get medical treatment in another area?

We clarify that there were vaccinations in Abu Dhabi prior to the study's start date. The choice of our study's start date was closely tied to the availability and completeness of patient records.

It is, however, unlikely that the inclusion of data from before the start date of our study would have substantially altered our findings. The vaccination campaign's early phase primarily targeted specific population groups, and the scale of vaccination during this period was relatively limited compared to the period covered in our study. Therefore, we believe our results are representative of the broader vaccine rollout's impact during the study period.

Regarding the possibility of individuals being vaccinated in Abu Dhabi and receiving medical treatment elsewhere, this is indeed a possibility. Individuals may seek medical care in different areas or even in other countries after vaccination and may be lost to follow-up post-vaccination. This aspect introduces a potential limitation to our study, and other similar studies, as it could lead to underreporting of adverse events or outcomes post-vaccination. We have acknowledged this limitation in our original manuscript and stress this limitation in the revised manuscript.

3.7. Line 166. 474 events in the risk period – what is the post hoc power to detect a 50% increase in risk with the composite outcome in this study. This is really just to give context to the negative results in the study.

The post-hoc power analysis, based on 474 events in the risk period (following 1.7 million exposures in total), indicated that we had an 84% power (at a level of significance of 0.05) to detect a 30% increase in risk with the composite outcome.

We included a description of the power analysis conducted, along with the results indicating the detectable effect size given our sample size. Please, refer to point 3.3 for a discussion on the power analysis performed.

3.8. Line 177. Can you please just clarify that there were 399 individuals but 474 admissions from these 399 individuals. What are the corresponding numbers from the comparison period.

We clarify that one individual may have multiple events (e.g., one individual can have an event after each dose of the vaccine). The corresponding numbers from the comparison period are 446 individuals and 532 events (as per Table 2).

3.9. Line 181. It is strange to give the dose 2 and dose 3 numbers without the dose 1 numbers

We have amended and added dose 1 numbers in the revised manuscript.

3.10. Line 181-186. You are saying that the event rate in the risk interval for both vaccines and for all 3 doses tends to be higher than the event rate in the comparative period, though it is mainly for dose 1 and dose 3. How much of this could be due to the imbalances in when individuals got vaccinated. All of the adjusted IRRs are below 1 for dose 3.

This observation highlights that unadjusted event rates in the risk interval tend to be higher than those in the comparison interval (comparing the first two columns of Table 3). This trend is particularly noticeable for the first and third doses. These are unadjusted event rates and do not account for various confounding factors. We then focus our analysis on the incidence rate ratios (IRRs) which are adjusted for potential confounders, which account for the confounding factors.

Regarding the Reviewer's question about the imbalances in the timing of vaccinations, we note that our observation is that individuals vaccinated at different times may have varying risk profiles due to several factors, including changes in vaccine distribution strategies, evolving public health guidelines, and shifts in the pandemic's dynamics.

We have added a minor amendment to the revised manuscript for clarifying that statement above concerns unadjusted event rates.

3.11. Table 3 there are 3 sets of identical values for dose 1 and dose 2 in the comparison interval which seems a bit unusual. The same thing happens in Table 5 so is this due to using exactly the same comparative group for dose 1 and dose 2?

The Reviewer's observation is correct. For dose 1 and 2, for the comparison interval, we selected vaccine recipients which may have been administered dose 1 or dose 2. Hence, in some cases, the same number of events was obtained.

3.12. Table 3. I am puzzled by the numbers and event rates. For dose one the event rates are similar 11.2 and 10.9 but the numbers of events are quite different 131 to 213 which means that the person time in the risk period must be a lot

less than the person time in the comparison interval, yet both are 21 days. If vaccine delivery was uniform, which it is unlikely to be, I would have expected person time to be similar in the two intervals. I see that this is probably due to comparing a dose 3 vaccinated person 1-21 days post vaccination with dose 1/2/3 vaccinated individuals (22-42 days post latest vaccine). I can see that you do this to increase precision but can you discuss the potential issues with this. For dose 3 which is largely given to older individuals the majority of their comparators will be much younger and you are then using the adjustment model. For dose 1 you will be comparing more similar type of individuals but potentially you could have a dose 2 comparator in the same analysis as a dose 1 vaccinated person. I would certainly like to see a sensitivity analysis where dose 1 vaccinated were compared to dose 1 controls only, same for dose 2 and 3.

The Reviewer's observations are correct. Despite similar event rates (e.g., 11.2 and 10.9), the number of events differs significantly (131 to 213). This discrepancy is indeed due to the difference in person-time between the risk and comparison intervals, which, while both spanning 21 days, do not have equivalent person-time due to variations in vaccine delivery and uptake. The dynamics of vaccine rollout, particularly in the initial phases, are often non-uniform, leading to such disparities in person-time between the intervals. We would note that in other studies (cited in the manuscript), conducted in other countries/regions, similar observations can be made.

Regarding the method of comparison, we chose to compare individuals in the 1-21 days post-vaccination (risk period) with those in the 22-42 days post-vaccination (comparison period) for all doses. While we understand the Reviewer's interest in a sensitivity analysis comparing dose 1 vaccinated individuals to dose 1 controls only, and similarly for doses 2 and 3, we have reservations about this approach. One significant concern is the reduction in sample size, which could lead to lower statistical power and potentially less reliable results. Another key challenge in conducting this analysis arises from the vaccination schedule itself, particularly for dose 2, which was typically administered within 3 to 4 weeks of dose 1. This close spacing significantly limits the number of individuals available for a meaningful comparison group for dose 1. Furthermore, in Abu Dhabi and the broader UAE, most of the population received a third dose, which makes the comparison groups (for dose 3) more similar than if only a specific group of people had received it (as pointed out by the Reviewer). Our chosen approach, which compares individuals across a broader timeframe, was specifically designed to increase the precision of our findings, considering the realities of the vaccine rollout schedule and uptake patterns. Given these considerations, we believe that the proposed sensitivity analysis, while theoretically interesting, would not add substantial value to our study.

3.13. In Table 3 the numbers of events for each of the doses adds up to 416. Can you please clarify this number. In the text you have reported 399 individual and 474 events. I can see that this could be due to vaccinated individuals having the composite event after different doses, in which case could you please clarify this in the methods. Currently in Lines 100-101 you have that the first hospitalisation on each day is used and on first read I (erroneously) took this to mean the first hospitalisation during the study period.

The Reviewer's observation is correct. Individuals can have multiple events, which may occur after dose 1 and dose 2, for example. We have added a minor amendment to the Results section to clarify this point.

3.14. Table 3. last row – the CI for the IRR goes from -0.73 to 0.01. I don't think that this can be correct. What method were you using for the CIs?

We thank the Reviewer for spotting this "typo". The correct interval is (-0.73 to 1.01), which has been amended corrected in the revised manuscript. 95% CIs were computed using the standard error derived from the Poisson regression models fitted to the data.

3.15. Lines 100-101. With the outcome definition I think that if someone has an ER visit for a condition 5 days after vaccination and an hospital admission on day 7 after the vaccination then this would count as 2 events. Many other studies of vaccine safety just look at the first event for the condition in the risk and comparison periods. The number of

individuals involved may not be great but I think you should discuss the potential magnitude of this issue.

The observation of the Reviewer is correct. From our analysis, only two individuals experience more than one event within a given risk/comparison period. Hence, we do not expect these to change our findings.

3.16. Table 5. The caption states primary outcome but these are secondary outcomes in the methods.

We thank the Reviewer for highlighting this. We have amended the caption in the revised manuscript.

3.17. Lines 217 -219. I find this sentence confusing.

We amended the sentence and made the statement clearer in the revised manuscript.

REVIEWER COMMENTS

Reviewer #1 (Remarks to the Author):

The manuscript was improved. I still have a few remaining comments:

1.1 The statement is much clearer. However, the last sentence of the abstract still seems confusing: does it only apply to Pfizer vaccine?

1.4 a few additional references that could have been considered

<https://doi.org/10.12997/jla.2023.12.2.119>, DOI: 10.7326/M22-0988, DOI: 10.1093/cvr/cvac068

1.5 Temporality can be accounted for in SCCS modelling, I'm not sure the following sentence is still an issue in such situation: "In a self-controlled design, outcome events in the comparison interval invariably occur on later calendar dates than those in the risk interval."

Furthermore, does the following sentence implies that we should not adjust for all time-invariant confounders? If yes, could the authors provide an example? "Moreover, the chosen approach allowed for a more nuanced adjustment for time-invariant confounders."

1.6 This part would benefit from more discussion on the differences. I may be wrong but I would have expected an association with inactivated vaccine, as with adenoviral-vectored vaccines

Reviewer #2 (Remarks to the Author):

Thank you for giving me the opportunity to review this paper again, and thank you to the authors for expanding on their methodology. Despite the authors clarification and the references provided, the study design used (case-only, exposure-anchored, and using a between-person comparison) is not one I'm familiar with, so I cannot comment on whether it's been appropriately implemented and will leave that to other reviewers. I do feel that expanding on the design within the paper itself would be useful: the VSD references all use a range of study designs, and not all of them appear to use case-only comparisons (i.e, 31).

My other questions have been largely addressed, if space, I would include the more detailed description of the data source in the paper itself as this is useful.

I have the following additional comments on the revised version:

- It is not accurate to state that comparison intervals in an SCCS always occur later than the risk intervals, usually time both before and after vaccination is included in the reference window, although variants of the SCCS such as the SCRI can be unidirectional in nature (with comparison windows either before or after the risk window). I'm also not sure what the authors mean that the chosen approach allowed for a "nuanced adjustment" of time-invariant confounders: surely the complete adjustment offered by a self-controlled design would be useful? I agree though, that the use of concurrent controls would be expected to control more effectively for calendar time trends as there is no need to specify their form.

- I found the added post-hoc power calculations quite helpful for interpreting some of the results, but was wondering whether the authors could clarify – in the paper - for which outcome incidences these were conducted? Would you still have had sufficient power for each individual outcome (confidence intervals in Table 5 suggests not?). I think lack of power to study each individual outcome warrants a slightly more cautious conclusion in the abstract and discussion (although it is reassuring that events were so rare!).

Reviewer #3 (Remarks to the Author):

Thanks for your revised manuscript and your explanations of you chosen method of analysis.

I am very happy with your rationale for the use of this study design in Abu Dhabi. Personally I

think that a self controlled analysis would work with the data that you have but am content that your method of analysis is equally valid. I think that while the matching of calendar date is a strength of your approach the reliance on statistical adjustments for age, sex etc is a relative weakness. Every statistician has to make informed judgements as to the most appropriate methods of analysis for their data.

Also thanks for looking into the post hoc power as this suggests that your study is adequately powered to pick up a 30% increase in rate ratios. I think that this is adequate.

I still think that there are areas where you could present more information in the supplementary materials to explain why this matching method is necessary in Abu Dhabi and why the self controlled case series is not. Your comments have implications for the many other covid vaccination safety studies carried out using the self controlled case series.

One of your main reason for using this study design is that there are strong patterns in the day to day variation in vaccine receipt and in hospital admission for adverse events. The adverse events seem to me to be serious requiring an emergency admission to hospital so am puzzled a bit by the day to day variations. Can you plot the temporal pattern in the composite outcome over the study period. I would find this a valuable plot - just for the composite outcome. Trends in vaccine administration, on the same temporal scale, by vaccine dose would also be helpful.

I think that your study is very similar to

Klein, N. P.; Lewis, N.; Goddard, K.; Fireman, B.; Zerbo, O.; Hanson, K. E.; Donahue, J. G.; Kharbanda, E. O.; Naleway, A.; Nelson, J. C.; Xu, S.; Yih, W. K.; Glanz, J. M.; Williams, J. T. B.; Hambidge, S. J.; Lewin, B. J.; Shimabukuro, T. T.; DeStefano, F.; Weintraub, E. S. Surveillance for Adverse Events After COVID-19 mRNA Vaccination. *JAMA* 326 (14), 1390-1399 (2021).

without the repeated testing. This study used very fine stratification for age group, sex, region and then averaging the stratified estimates of being in the risk interval compared to the comparator interval. I think that you appear to have used adjustment. Can you clarify this and include in the supplementary materials the estimated coefficients from the models.

I am glad that you clarified that each person is only in a comparator period after receipt of their last dose and this explains the same numbers in the tables. Does this mean that the majority of the comparator periods come from individuals who had 3 doses? And does this mean they tend to be older and so the age adjustment is particularly important in this analysis. The information in Table 1 suggests that for both vaccines the % in the 40-64 age group is much larger for dose 3 compared to doses 1 and 2 (almost twice as large). Could you provide a table in the supplementary material describing the individuals who are in the comparator periods

Table 1:

What does the Only 1 dose row mean? I am puzzled that under 30% of those with 1 dose of either vaccine have one dose only but over 95% of those with 3 doses have one dose only. I think that this needs to be clarified.

Response to Reviewer's Comments

We have provided point-by-point responses to all the comments from the reviewers. We have retained the original comments and statements and marked our responses in red to make the revision process easier for reviewers.

Comments from reviewer #1

The manuscript was improved. I still have a few remaining comments:

1.1 The statement is much clearer. However, the last sentence of the abstract still seems confusing: does it only apply to Pfizer vaccine?

We thank the Reviewer for this note. No, it applies to both vaccines. We have amended the Abstract accordingly (and we replicate it below):

"This study found no substantial link between receiving either mRNA and inactivated COVID-19 vaccines and a higher likelihood of cardiovascular or hematological events within 21 days after vaccination. "

1.4 a few additional references that could have been considered <https://doi.org/10.12997/jla.2023.12.2.119>, DOI: 10.7326/M22-0988, DOI: 10.1093/cvr/cvaco68

We acknowledge the relevance of the references suggested by the Reviewer to our study.

[1] Chen, C. Y.; Su, T. C. Benefits and Harms of COVID-19 Vaccines in Cardiovascular Disease: A Comprehensive Review. *J Lipid Atheroscler.* 12 (2):119-131 (2023).

The authors reviewed multiple studies and case reports on cardiovascular complication following the administration of COVID-19 vaccines. The authors included multiple studies, some of which are also cited in our study. They concluded that, from the different CVD complications, most evidence focuses on myocarditis and pericarditis, which are most strongly associated with mRNA vaccines and predominantly occur in young males within days of receiving the second dose; although claiming that evidence is still scarce, and that further studies are needed to understand the associations found between COVID-19 vaccination and CVD-related complications including myocardial infarction, arrhythmia, and stress cardiomyopathy.

[2] Botton, J.; Jabagi M. J.; Bertrand, M.; Baricault, B.; Drouin, J.; Le Vu, S.; Weill, A.; Farrington, P.; Zureik, M.; Dray-Spira, R. Risk for Myocardial Infarction, Stroke, and Pulmonary Embolism Following COVID-19 Vaccines in Adults Younger Than 75 Years in France. *Annals of Internal Medicine* 175 (9): 1250–57 (2022).

In this study conducted in France, which included subjects younger than 75 years, the first dose of ChAdOx1 was found to be associated with myocardial infarction and pulmonary embolism in the second week of vaccination; in addition, the study found an increase in the number of hospitalizations associated with myocardial infarction in the second week after a single dose of the Ad26.COV 2.S vaccine. This study adds to the body of literature found (and included in our discussion) with similar findings.

[3] Ye, X.; Ma, T.; Blais, J. E.; Yan, V. K. C.; Kang, W.; Chui, C. S. L.; Lai, F. T. T.; Li, X.; Wan, E. Y. F.; Wong, C. K. H.; Tse, H. F.; Siu, C. W.; Wong, I. C. K.; Chan, E. W. Association between BNT162b2 or CoronaVac COVID-19 vaccines and major

adverse cardiovascular events among individuals with cardiovascular disease. *Cardiovasc Res.* 118 (10): 2329-2338 (2022).

The study above was conducted in Hong Kong and found that among individuals with a prior history of CVD there is no evidence of an increased risk of major adverse cardiovascular events after the BNT162b2 vaccine (nor CoronaVac), which is in agreement with the findings from our subgroup analyses.

Accordingly, we have made minor amendments to our manuscript and incorporated these references to enrich the body of literature cited (Introduction and Discussion sections). This addition allows for a more comprehensive comparison of existing findings with our own.

1.5 Temporality can be accounted for in SCCS modelling, I'm not sure the following sentence is still an issue in such situation: "In a self-controlled design, outcome events in the comparison interval invariably occur on later calendar dates than those in the risk interval."

Furthermore, does the following sentence implies that we should not adjust for all time-invariant confounders? If yes, could the authors provide an example? "Moreover, the chosen approach allowed for a more nuanced adjustment for time-invariant confounders."

The Reviewer's point about SCCS models being capable of accounting for temporality is recognized. Indeed, SCCS can adjust for time-varying confounders within the framework by, for example, designating different time periods as different strata in the analysis. However, our concern, as noted in our manuscript, relates specifically to the dynamics of vaccine rollout and health event surveillance where temporal associations could change very quickly. In our study context, events within the risk period and the subsequent comparison period could be influenced by rapidly changing external factors such as policy shifts or public health interventions (as it was the case in many countries), which might not be directly accounted for with SCCS approaches. Our statement highlighted the inherent lag between the risk and comparison intervals in traditional SCCS designs, which might not capture these rapid changes effectively.

The sentence regarding the nuanced adjustment for time-invariant confounders was intended to reflect the capabilities of our chosen method in addressing confounders that do not change over time within an individual but might have different impacts depending on the external context. SCCS inherently controls for time-invariant confounders such as chronic health conditions because it uses each individual as their own control. However, our approach allowed us to model the interaction more flexibly between these time-invariant characteristics and time-varying external influences. For example, an individual's underlying health condition (a time-invariant confounder such as a chronic condition) might interact differently with a vaccine's effectiveness or side effects depending on the timing within the vaccine rollout (a time-varying external factor). This allowed us to account more effectively for the interactions between constant individual characteristics and the dynamic external conditions, crucial for accurately assessing risks during a public health crisis.

To summarize, while SCCS is a robust method for controlling for many types of confounders, we chose an approach that could more flexibly handle the rapid changes and complex interactions typical of public health interventions (such as lockdowns and movement restrictions) during a pandemic. The selected method for our study was chosen not because SCCS is ineffective in general, but because we wanted to attempt to capture more accurately the dynamics at play and carefully balance the need for controlling calendar time effects with the requirement to account for individual-level and systemic variations. We hope this clarifies the choices made in our study design and the statements made in our manuscript.

1.6 This part would benefit from more discussion on the differences. I may be wrong but I would have expected an association with inactivated vaccine, as with adenoviral-vectored vaccines

We recognize the extensive research on the BNT162b2 vaccine and aimed to contribute additional insights tailored to our specific population and methodology. Our analysis of the BBIBP-CorV vaccine, in particular, is designed to provide a nuanced comparison with other inactivated vaccines, filling a gap in existing literature by examining its safety with respect to cardiovascular and hematology-based events on a diverse population.

We have expanded our discussion in the revised manuscript to compare the safety profiles of BBIBP-CorV more robustly with other inactivated vaccines, and indeed, with the ChAdOx1 vaccine. This enhanced discussion is based on a thorough review of current literature and our specific study findings. This approach not only addresses the gap the Reviewer noted but also enriches the discourse on vaccine safety across different platforms.

Comments from reviewer #2

Thank you for giving me the opportunity to review this paper again, and thank you to the authors for expanding on their methodology. Despite the authors clarification and the references provided, the study design used (case-only, exposure-anchored, and using a between-person comparison) is not one I'm familiar with, so I cannot comment on whether it's been appropriately implemented and will leave that to other reviewers. I do feel that expanding on the design within the paper itself would be useful: the VSD references all use a range of study designs, and not all of them appear to use case-only comparisons (i.e, 31).

My other questions have been largely addressed, if space, I would include the more detailed description of the data source in the paper itself as this is useful.

Thank you for the Reviewer's constructive comments. We have already extended other sections of the paper, including the discussion, and have made a minor addition to the dataset description. While we are unsure what further details to expand upon in this section, we have included references to other studies that also describe the dataset to provide additional context.

I have the following additional comments on the revised version:

- It is not accurate to state that comparison intervals in an SCCS always occur later than the risk intervals, usually time both before and after vaccination is included in the reference window, although variants of the SCCS such as the SCRI can be monodirectional in nature (with comparison windows either before or after the risk window). I'm also not sure what the authors mean that the chosen approach allowed for a "nuanced adjustment" of time-invariant confounders: surely the complete adjustment offered by a self-controlled design would be useful? I agree though, that the use of concurrent controls would be expected to control more effectively for calendar time trends as there is no need to specify their form.

The Reviewer is correct in that standard SCCS designs often include reference windows both before and after the risk window, and we appreciate your clarification regarding the omnidirectional nature of some variants. Our statement aimed to highlight the specific limitations we observed with standard SCCS in having concurrent comparison periods as used in our study design. We have amended this statement in the manuscript.

Regarding the adjustment of time-invariant confounders, the methodology used does indeed differ from the complete adjustment inherent in self-controlled designs. We employed a method that controls for these confounders while

allowing for a flexible examination of how these interact with dynamic external conditions over the study period. We agree with the Reviewer on the benefit of using concurrent controls to more effectively manage calendar time trends without needing to specify their form explicitly.

- I found the added post-hoc power calculations quite helpful for interpreting some of the results, but was wondering whether the authors could clarify – in the paper - for which outcome incidences these were conducted? Would you still have had sufficient power for each individual outcome (confidence intervals in Table 5 suggests not?). I think lack of power to study each individual outcome warrants a slightly more cautious conclusion in the abstract and discussion (although it is reassuring that events were so rare!).

We thank the Reviewer for acknowledging the value of the post-hoc power calculations in our study.

The power calculations were specifically performed for the primary composite outcome, which we have now detailed in the manuscript for clarification. We recognize the Reviewer's concern regarding the power for each individual outcome. Indeed, as suggested by the confidence intervals in Table 5, our study did not have sufficient power for analyzing each individual outcome independently. We have addressed this in the discussion section, acknowledging it as a limitation of our study, which reinforces the need for a more cautious interpretation of the findings (as discussed).

Comments from reviewer #3

Thanks for your revised manuscript and your explanations of your chosen method of analysis.

I am very happy with your rationale for the use of this study design in Abu Dhabi. Personally I think that a self controlled analysis would work with the data that you have but am content that your method of analysis is equally valid. I think that while the matching of calendar date is a strength of your approach the reliance on statistical adjustments for age, sex etc is a relative weakness. Every statistician has to make informed judgements as to the most appropriate methods of analysis for their data.

Also thanks for looking into the post hoc power as this suggests that your study is adequately powered to pick up a 30% increase in rate ratios. I think that this is adequate.

We thank the Reviewer for acknowledging the value of the post-hoc power calculations in our study and for raising and suggesting its application in the first place.

I still think that there are areas where you could present more information in the supplementary materials to explain why this matching method is necessary in Abu Dhabi and why the self controlled case series is not. Your comments have implications for the many other covid vaccination safety studies carried out using the self controlled case series.

One of your main reasons for using this study design is that there are strong patterns in the day to day variation in vaccine receipt and in hospital admission for adverse events. The adverse events seem to me to be serious requiring an emergency admission to hospital so am puzzled a bit by the day to day variations. Can you plot the temporal pattern in the composite outcome over the study period. I would find this a valuable plot - just for the composite outcome. Trends in vaccine administration, on the same temporal scale, by vaccine dose would also be helpful.

We agree with the Reviewer that additional visualizations of the temporal patterns in vaccine receipt and hospital admissions would indeed provide valuable insights. In response, we have included such plots in the revised manuscript (in supplementary material) to illustrate the observed day-to-day variations in both vaccine administrations and (composite) outcome events.

It is important to note that these variations are influenced not only by vaccine administration schedules (e.g., rollout campaigns) but also by broader (external) time-varying factors (such as lockdowns, movement restrictions) during the pandemic. These factors can significantly affect both vaccine uptake and the incidence of hospital admissions, which underscores the need for flexible analytical approaches. We acknowledge the effectiveness of SCCS designs, as evidenced by their widespread use in epidemiological studies, and we are not suggesting they were not a suitable choice. However, we firmly believe that our decision to use concurrent comparators is justified, as it also provides a suitable and robust framework for addressing the complex, rapidly changing environment of a global pandemic. We would argue that the chosen approach allowed us to more flexibly model the interaction between these time-invariant characteristics and time-varying external influences. For example, an individual's underlying health condition (a time-invariant confounder such as a chronic condition) might interact differently with a vaccine's effectiveness or side effects depending on the timing within the vaccine rollout (a time-varying external factor). The selected method for our study was chosen not because SCCS is ineffective in general, but because we wanted to attempt to capture more accurately the dynamics at play and carefully balance the need for controlling calendar time effects with the requirement to account for individual-level and systemic variations.

We trust that the enhancements made to the manuscript and our rationale for the chosen methodology address the Reviewer's concerns, affirming its appropriateness for our study. This methodological choice, we argue, should not just be seen as an alternative but as a complementary methodology that can be used for the broader context of vaccine safety and surveillance (as used in vaccine-safety studies).

I think that your study is very similar to

Klein, N. P.; Lewis, N.; Goddard, K.; Fireman, B.; Zerbo, O.; Hanson, K. E.; Donahue, J. G.; Kharbanda, E. O.; Naleway, A.; Nelson, J. C.; Xu, S.; Yih, W. K.; Glanz, J. M.; Williams, J. T. B.; Hambidge, S. J.; Lewin, B. J.; Shimabukuro, T. T.; DeStefano, F.; Weintraub, E. S. Surveillance for Adverse Events After COVID-19 mRNA Vaccination. *JAMA* 326 (14), 1390-1399 (2021).

without the repeated testing. This study used very fine stratification for age group, sex, region and then averaging the stratified estimates of being in the risk interval compared to the comparator interval. I think that you appear to have used adjustment. Can you clarify this and include in the supplementary materials the estimated coefficients from the models.

The Reviewer is correct; while Klein et al. employed a fine stratification for age group, and other demographics and then averaged the stratified estimates of being in the risk interval compared to the comparator interval, we opted for a model that adjusts for these factors (as described in our Methods section). We believe this adjustment approach allowed us to maintain robustness while efficiently handling the complexities of the data. While we have chosen not to include all coefficients from all models, we believe the manuscript provides sufficient methodological clarity to understand the adjustments made and the rationale behind the approach.

We hope this explanation addresses the Reviewer's query and clarifies the distinctions between our methodology and that of the referenced study. These can be made available in future in line with the data and code availability statements.

I am glad that you clarified that each person is only in a comparator period after receipt of their last dose and this explains the same numbers in the tables. Does this mean that the majority of the comparator periods come from individuals who had 3 doses? And does this mean they tend to be older and so the age adjustment is particularly important in this analysis. The information in Table 1 suggests that for both vaccines the % in the 40-64 age group is much larger for dose 3 compared to doses 1 and 2 (almost twice as large). Could you provide a table in the supplementary material describing the individuals who are in the comparator periods

To address the Reviewer's query: it is not the case that the majority of the comparator periods predominantly come from individuals who have received three doses. Instead, depending on the calendar date, the comparator periods are distributed across individuals who received varying numbers of doses, not skewed towards those who received three doses (except halfway through the study period).

Regarding the age distribution, while it is observed that a higher percentage of individuals in the 40-64 age group received the third dose compared to the first two doses, this does not imply that older individuals predominately constitute the comparator periods. The age adjustment in our analysis also ensures that any potential confounding due to age distribution across different dose groups is appropriately managed. We appreciate the suggestion to include additional descriptive information about individuals in the comparator periods. We believe the current manuscript and supplementary materials provide sufficient detail on the study population and the structuring of comparator periods. Further breakdowns by individual dose groups and specific demographic details maintain the clarity and focus of our analysis without overcomplicating the presentation of our findings (e.g., we note the same individual may be included as part of the comparator and risk groups at different calendar dates, which could complicate the interpretation of the results).

We hope this clarifies the Reviewer's concerns and assures that the analysis accounts for these factors effectively.

Table 1:

What does the Only 1 dose row mean? I am puzzled that under 30% of those with 1 dose of either vaccine have one dose only but over 95% of those with 3 doses have one dose only. I think that this needs to be clarified.

The row shows the proportion of individuals that were included in the study and only had one dose during the study period. We note that these observations can be explained by the fact that the study period began at a time when the vaccine rollout was already in progress; i.e., the study period did not include the first first doses given to (some) individuals. The third dose was given as a booster to individuals sometime (typically 6 to 8 months) after the administration of the first two doses, and they occurred during the study period. We further clarify that the commencement date of the study was determined based on the completeness of the available data.

We added this clarification to the manuscript (Results section).